# School-based screening for childhood anxiety problems and intervention delivery: a codesign approach

Victoria Williamson [1,2,3] Michael Larkin,[4] Tessa Reardon,[2,3] Samantha Pearcey,[2] Roberta Button,[2,3] Iheoma Green,[2,3] Claire Hill,[3] Paul Stallard,[5] Susan H Spence,[6] Maria Breen,[7] Ian Mcdonald,[8] Obioha Ukoumunne [9] , Tamsin Ford,[10] Mara Violato,[11] Falko Sniehotta,[12] Jason Stainer,[13] Alastair Gray [11] , Paul Brown,[14] Michelle Sancho,[15] Fran Morgan,[16] Bec Jasper,[16] Cathy Creswell[2,3]

For numbered affiliations see end of article.

**Correspondence to**
Dr Michael Larkin;
m.larkin@aston.ac.uk

## ABSTRACT

**Objectives** A very small proportion of children with anxiety problems receive evidence-based treatment. Barriers to access include difficulties with problem identification, concerns about stigma and a lack of clarity about how to access specialist services and their limited availability. A school-based programme that integrates screening to identify those children who are most likely to be experiencing anxiety problems with the offer of intervention has the potential to overcome many of these barriers. This article is a process-based account of how we used codesign to develop a primary school-based screening and intervention programme for child anxiety problems.

**Design** Codesign.

**Setting** UK primary schools.

**Participants** Data were collected from year 4 children (aged 8–9 years), parents, school staff and mental health practitioners.

**Results** We report how the developed programme was experienced and perceived by a range of users, including parents, children, school staff and mental health practitioners, as well as how the programme was adapted following user feedback.

**Conclusions** We reflect on the mitigation techniques we employed, the lessons learnt from the codesign process and give recommendations that may inform the development and implementation of future school-based screening and intervention programmes.

## INTRODUCTION

Anxiety disorders are among the most prevalent mental health disorders experienced by children, with 6.5% of children globally meeting likely diagnostic criteria[1] and as many as half of lifetime anxiety disorders starting before a child leaves primary school.[2] Without intervention, anxiety disorders can persist into adulthood with deleterious implications for a child's social, educational and familial functioning.[3]

Effective treatments, such as cognitive behaviour therapy (CBT),[4] have been

## STRENGTHS AND LIMITATIONS OF THIS STUDY

⇒ The codesign methodology used allowed for the collection of data from a broad range of users (parents, children, teachers, practitioners) at various stages of the study, providing in-depth insight into their experiences and concerns at each research stage.

⇒ Our use of codesign also yielded a number of transferrable learning points that may be applicable to other studies aiming to implement universal mental health screening and intervention in schools.

⇒ The inclusion of a range of participant perspectives highlighted that some school staff and practitioners may have very different views from families about the potential risks and benefits to a school-based mental health screening/intervention pathway.

developed for childhood anxiety disorders, yet only a small proportion of children with anxiety disorders actually access services at all, let alone evidence-based treatment.[5 6] Barriers include problems with identification and difficulties in accessing treatment, including parental concerns about children being labelled or families blamed for child difficulties; a lack of confidence or ability to identify likely child anxiety problems among primary care providers, school staff or other professionals that children interact with; parental uncertainty about how to find reliable sources of support; and restricted access to specialist services due to narrow inclusion criteria or long waiting lists.[3 5 7 8] A school-based screening programme to identify children who are most likely to be experiencing anxiety problems and offer intervention seamlessly without families having to negotiate routes to services has the potential to overcome many of these barriers. However, if carried out poorly school-based screening programmes may also have poor uptake or inadvertent unintended consequences, such

as increasing stigma or misidentification.[9 10] Designing engaging, acceptable and well-received procedures is therefore essential.

For such a programme to be implemented, it must function efficiently, be safe and reliable and have the experiences of service users and stakeholders at the heart of programme design and delivery.[11] This final criterion is best met by codesign—a method which aims to develop a thorough understanding of how stakeholders and service users perceive and experience the look, feel and procedures of a service which is then used to inform the design and delivery of and adaptations to services.[12] This approach brings advantages over surveys or questionnaires of patient/stakeholder experiences of a service as it allows for an in-depth understanding of a service's potential shortcoming and/or the development of solutions. A codesign approach allows for both participant views as well as patient and public involvement (PPI) perspectives to be incorporated, ensuring services are designed *for* users *with* users.[13] Codesign has been widely used in health contexts to make services more acceptable and, thus, ultimately improve patient well-being.[14–16] In relation to designing and delivering mental health services for children, previous qualitative codesign studies have yielded promising findings when the views of children, family members, clinicians and other stakeholders were incorporated.[17–19] Designing and implementing a successful school-based screening and intervention programme for childhood anxiety disorders requires equally thorough triangulation.

Our aim was to codesign an engaging and accessible primary school-based pathway to screen and offer an intervention for child anxiety problems. As potential screening tools[20] and low intensity interventions[21 22] already exist, the purpose of this study was to develop an in-depth understanding of the challenges that may arise when delivering screening and intervention for child anxiety problems in primary schools and to respond to such concerns by cocreating, implementing and evaluating solutions. In this article, we will provide a process-based account of how our school-based screening and intervention pathway was codesigned, how the pathway procedures were experienced by users and how pathway development was influenced by user feedback. We will

also report qualitative findings from interviews with parents, children, school staff and other stakeholders to show how their perspectives were incorporated in order to help ensure that the developed pathway would be well received and sustainably implemented.

## METHOD
### Approach and focus
We set out to codesign, produce and deliver a series of procedures—a 'pathway'—to improve access to an evidence-based intervention for child anxiety problems through primary schools in England. As described in detail in our study protocol,[23] several of the pathway features were specified in advance of the codesign work with input and guidance from stakeholder members of the research team (see next section). In particular, we prespecified that children's anxiety problems would be screened using validated questionnaire measures,[20] parents would receive feedback on the outcome and, where indicated, a brief online treatment for child anxiety problems would be offered. The treatment offered was an online version of a brief therapist-guided parent-delivered CBT approach for child anxiety problems (online support and intervention (OSI) for child anxiety) which involves seven online modules for parents, supported by a weekly 20 min telephone call with a children's well-being practitioner (CWP (psychological therapists with a 1-year postgraduate training), NHS Band 5[24]), with a follow-up telephone session 4 weeks later. A face-to-face version of this brief parent-led treatment has been found to be both clinically effective[25] and more cost-effective than an alternative brief psychological intervention[26]

As described in our protocol,[23] the codesign process to establish how the prespecified features of the pathway should be presented consisted of four stages. The first stage involved initial interviews and focus groups with parents, children, school staff and other stakeholders to inform the development of a set of procedures that would comprise the pathway (Stage 1) (see figure 1 and table 1). These procedures were subsequently applied in three primary schools (Stage 2) with participating children, parents and school staff providing feedback on their experience (Stages 3 and 4), including cued-recall interviews

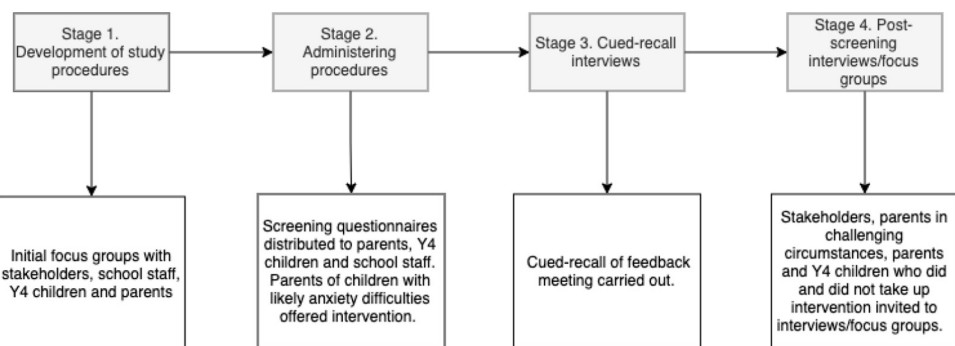

**Figure 1** Overview of the codesign process for developing the school-based screening and intervention pathway. Y4, year 4.

**Table 1** Stages 2–4 school demographic characteristics

| School | Total number of pupils on roll | Percentage of pupils with SEN support | Percentage of pupils eligible for free school meals | Percentage of pupils with English as an additional language |
|---|---|---|---|---|
| School 1 | 200 | 9.5% | 12.6% | 41.5% |
| School 2 | 364 | 18.0% | 9.1% | 23.9% |
| School 3 | 415 | 7.6% | 2.7% | 26.2% |
| National average | N/A | 12.2% | 20.8% | 19.2% |

National average refers to official UK government statistics for the 2020/2021 school year.[39]
SEN, special educational needs.

which examined parents' experiences of receiving feedback on whether their child experiences difficulties with anxiety (see table 2).

## PPI and stakeholder involvement

Parents, school staff and other stakeholders were involved in this codesign study in a number of ways. First, this project actively involved a dedicated PPI and stakeholder group from the protocol development stage to ensure that the developed pathway would be acceptable to both parents and school staff. This group included two parents with relevant lived experience as a parent of a child with anxiety problems, two school leaders and one school mental health lead for a national charity. Our PPI/stakeholder group provided guidance during the initial project plans and funding application and later informed the development of the study protocol and reviewed research data collected throughout the study to aid in decision-making. Examples of decisions that were made on the basis of consultation with this group included providing the option for children to complete screening measures at home (Stage 2), as well as guiding the researcher team on what information had to be securely shared about participating families with school staff for safeguarding purposes. Researchers met with the PPI/stakeholder group at regular intervals and the group were compensated for their time and expertise. The dedicated PPI/stakeholder group participants, while providing guidance, were not research participants. The dedicated PPI/stakeholder group were not directly involved in the recruitment of participants. Second, a distinct online PPI group, made up primarily of parents, was established for this project. Regular updates about the study as well as polls and questions were posed to the online PPI group in order to access wider parental views about study procedures and gain insight about key concerns. Results will be disseminated to participants via social media and lay summaries.

## Participants
### Sampling rationale for the codesign activities

For Stages 1–4, participants included children in year 4 of primary school (Y4; aged 8–9 years), parents of Y4 children, primary school staff and other stakeholders (see table 2). Y4 children (aged 8–9 years) were the focus of

the intervention as consultations with parents and school-staff advised that this would be a manageable time for primary schools. The delivery of the procedures in Y4 was thought to allow primary schools to see the benefit of the pathway and would enable children to thrive when managing subsequent key transitions (eg, to secondary school).

## Setting

Participants for Stage 1 were recruited from two local mainstream primary schools as well as through adverts online on social media and national mailing lists for the initial procedure development phase (see figure 1; Stage 1). Three local primary schools participated in Stages 2–4 to iteratively try out and adapt the pathway procedures (one school from Stage 1 and two new schools). These schools varied in their demographic characteristics (see table 1)

## Recruitment to the codesign activities
### Parents and children

To recruit participants with a broad range of perspectives to Stage 1, we circulated study invitations to families of all Y4 children in two primary schools in the local area, as well as circulating study adverts online on social media, and national mailing lists. In Stages 2–4, study information was circulated to all Y4 parents and children in three participating schools, including invitations to take part in the screening/intervention pathway and the opportunity to participate in study-related interviews. All Y4 parents and children in participating schools were invited to participate and were included in the study if they provided informed consent/assent.

Notably, in Stage 4, we also specifically recruited a number of parents facing challenging circumstances that could influence their views of the acceptability of and likely engagement with a school-based screening and intervention programme. These were parents who care for a foster child or a child with chronic physical health problems, where the parent has past/present mental health problem(s) or where the parent is a member of the UK Armed Forces community. This subgroup of parents (n=10, see table 2) was recruited via circulation of study advertisements online and via mailing lists. Parents who expressed an interest in taking part were approached

**Table 2** Overview of codesign input sources and data contributions

| Input | Participants | Study stage | Time frame | Demographic information | | Mode of contribution to codesign | Output generated |
|---|---|---|---|---|---|---|---|
| PPI/stakeholder group (headteachers×2; parents×2; voluntary/community sector mental health in schools expert) | N=5 | Stages 1–4 | Pre study—month 12 | Age, mean (SD) | 49.3 (7.4) | Regular meetings to share findings and discussion of study progress. | The dedicated PPI/stakeholder group members are part of the research team and provided guidance and recommendations on study findings and developments. |
| | | | | Females (n) | 2 | | |
| Practitioners who provide mental health support in schools | N=2 | Stage 1 | Months 1–2 | Age, mean (SD) | 54.5 (12.0) | Focus group interview conducted face to face (qualitative). | Perceptions of how the screening/intervention procedures should be introduced in schools, delivered, concerns and possible solutions. |
| | | | | Females (n) | 2 | | |
| | N=15 | Stage 4 | Months 6–8 | Age, mean (SD) | 38 (10.5) | Semistructured interviews conducted remotely (qualitative). | Perceptions of how the screening/intervention procedures should be introduced to the class, carried out, concerns and possible solutions. |
| | | | | Females (n) | 14 | | |
| Y4 children | N=8 | Stage 1 | Months 1–2 | Age range, years | 8–9 | Focus group interview conducted face to face (qualitative). | Identification of children who are likely to have problems with anxiety. |
| | | | | Females (n) | 6 | | |
| | N=29 | Stage 2 | Months 2–6 | Age, mean (SD) | 8.5 (0.6) | Completed screening questionnaires for likely anxiety problems (quantitative). | Experience of the screening pathway and intervention. |
| | | | | Females (n) | 19 | | |
| | N=2 | Stage 4 | Months 6–8 | Age, mean (SD) | 9 (0) | Semistructured interviews conducted remotely (qualitative). | Perceptions of how the screening/intervention procedures should be introduced to families, delivered in schools, concerns and possible solutions. |
| | | | | Females (n) | 2 | | |
| Y4 parents | N=7 | Stage 1 | Months 1–2 | Age, mean (SD) | 43.7 (3.6) | Focus group interview conducted face to face (qualitative). | Identification of children who are likely to have problems with anxiety. |
| | | | | Females (n) | 6 | | |
| | N=29 | Stage 2 | Months 2–6 | Age, mean (SD) | 42.0 (3.4) | Completed screening questionnaires for likely anxiety problems (quantitative). | Experience of the screening pathway and receiving feedback on scores. |
| | | | | Females (n) | 24 | | |
| | N=2 | Stage 3 | Months 5–6 | Age, mean (SD) | 46.5 (0.7) | Cued-recall interviews conducted via telephone (qualitative). | Experience of the screening pathway and intervention. |
| | | | | Females (n) | 2 | | |
| | N=7 | Stage 4 | Months 6–8 | Age, mean (SD) | 43.6 (2.2) | Semistructured interviews conducted remotely (qualitative). | Experience of the screening pathway and intervention. Includes parents who dropped out (n=2). |
| | | | | Females (n) | 6 | | |

Continued

**Table 2** Continued

| Input | Participants | Study stage | Time frame | Demographic information | | Mode of contribution to codesign | Output generated |
|---|---|---|---|---|---|---|---|
| Parents in challenging circumstances | N=10 | Stage 4 | Months 5–12 | Age, mean (SD) | 47.1 (7.6) | Semistructured interviews conducted remotely (qualitative). | Perceptions of how a school screening/intervention pathway could be delivered in schools and possible barriers/facilitators to taking part. |
| | | | | Females (n) | 7 | | |
| School staff | N=6 | Stage 1 | Months 1–2 | Age, mean (SD) | 48.0 (7.4) | Focus group interview conducted face to face (qualitative). | Perceptions of how the screening/intervention procedures should be introduced, delivered, concerns and possible solutions. |
| | | | | Females (n) | 6 | | |
| | N=4 | Stage 2 | Months 2–6 | Age, mean (SD) | 41.8 (8.3) | Screening questionnaires for likely anxiety problems (quantitative). | Identification of children who are likely to have problems with anxiety. |
| | | | | Females (n) | 2 | | |
| | N=5 | Stage 4 | Months 6–9 | Age, mean (SD) | 41.6 (7.2) | Semistructured interviews conducted remotely (qualitative). | Experience of the screening pathway, perceptions of the intervention offered to families and perceived barriers/facilitators to uptake in schools. |
| | | | | Females (n) | 3 | | |

PPI, patient and public involvement; Y4, year 4.

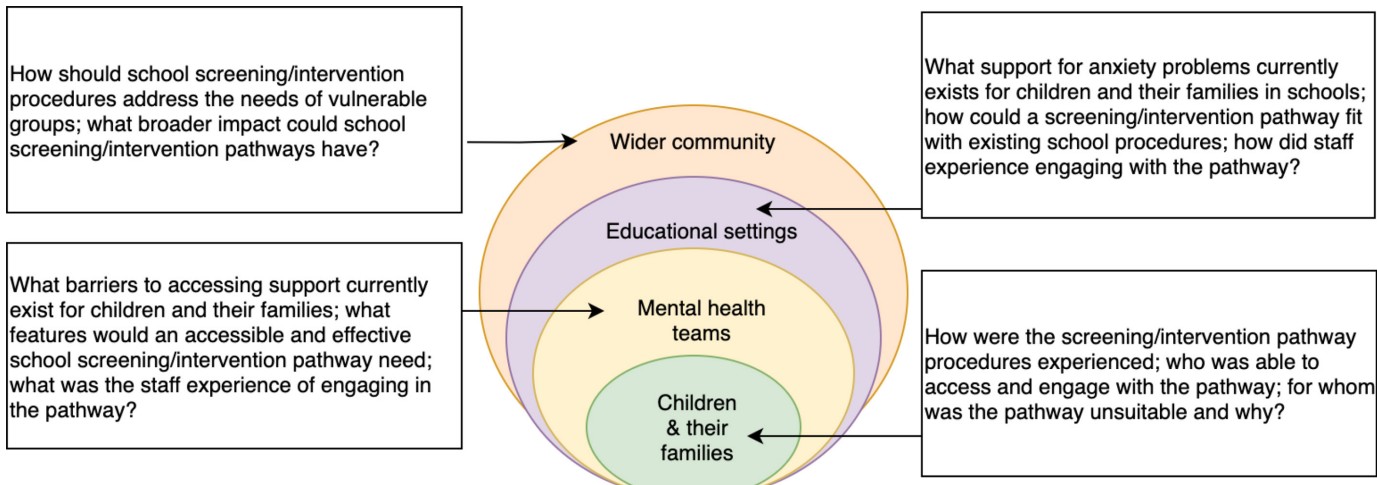

**Figure 2** Levels of investigation in codesign process.

by the research team, screened against study inclusion/ exclusion criteria and invited to take part following informed consent. The inclusion of this subgroup of parents aimed to ensure that the codesigned school-based programme would be inclusive and appropriate to the needs of a greater number of families (see Williamson *et al*, under review).

### School staff and other stakeholder participants

To recruit school staff and practitioners who provide mental health support in schools to Stages 1 and 4, we circulated invitations for study interviews/focus groups within local primary schools and shared study adverts online and via mailing lists. School staff and practitioners were encouraged to contact the research team if they were interested in taking part. School staff were included in study interviews if they were employed in a participating mainstream primary/junior school in England (eg, class teacher, headteacher). The inclusion criteria for staff that provide mental health support in schools were that they must be a practitioner providing mental health support in primary schools in England, such as educational psychologists, special educational needs coordinator and emotional literacy support assistants (Stages 1 and 4, see figures 1 and 2). For clarity, they are referred to throughout this manuscript as 'practitioners'. Practitioners were sampled to ensure that a range of views was represented from a diverse group of professional backgrounds and qualifications.

### Procedure and description of codesign process

Our codesign consultations were conducted throughout all four stages (see figure 1), to allow us to get feedback on a preliminary pathway prototype, refine it, implement it and then get feedback on people's experiences and perceptions of that to inform a further refinement.

### Stage 1

We carried out in-depth one-to-one interviews and focus groups with practitioners, school staff, children and

parents (see table 2 and online supplemental file 1). Participants were asked for their views on features of the draft pathway which the research team had outlined in collaboration with the dedicated PPI/stakeholder group's input. Participants were shown visual materials of the proposed stages of the pathway (when possible, if the interview was conducted in person or via video conference with the visuals representing the general journey through screening to intervention). The visuals were intended as a generic prototype of the pathway stages (ie, a generic image of a school was shown during questions about the potential impact screening may have on a school community) and participants were encouraged to write down further thoughts, comment on concerns and highlight possible solutions. When shown the pathway visuals, participants were asked about their beliefs about using screening questionnaires to identify child anxiety problems in schools, perceptions of how families should be informed of the outcomes of the screening questionnaires, families experiences of the online intervention and views of whether there might be any secondary effects of a school-based screening and intervention on a family or school community (see figure 2).

### Stage 2

The detailed prototype set of procedures refined after Stage 1 were administered in three primary schools, including screening, feedback to parents and the offer of treatment where indicated.

### Stage 3

Parents were invited to discuss their experience of receiving feedback on their child's screening outcomes via cued-recall interviews. The cued-recall interviews were audio recorded and transcribed verbatim. The aim of the cued-recall interviews was to capture the acceptability of the feedback procedures used here to inform a further iteration of the procedures ahead of a larger scale future trial. Participating parents received feedback on their children's

screening outcomes in writing and via telephone call from the study CWP. Recordings of the parent–CWP telephone call were reviewed by parents with a study researcher, with the parent encouraged to comment at points that were relevant, for example, points in the call where the parent felt more information from the CWP would have been useful.

### Stage 4
Following the administration of all the pathway procedures, interviews were carried out with Y4 children, their parents and school staff. We carried out interviews with a subsample of participating parents and children who completed the screening questionnaires and engaged with the treatment modules and of parents and children who withdrew. All parents who engaged with or withdrew from treatment were invited to interview. Parent interviews differed from the cued-recall interviews (Stage 3) in that parents were asked about their overall experience of the screening/intervention.

School staff in participating schools were interviewed about their experience of facilitating the pathway procedures. Practitioners who provide mental health treatment in primary school settings were also interviewed about their views of the pathway procedures that had been administered. Views about the proposed pathway were also sought from parents in especially challenging circumstances (eg, foster families, military families) (Williamson *et al*, under review). The interviews were used to gain an in-depth understanding of participants' experiences and perceptions of the pathway procedures. Participants' feedback and recommendations at this stage will inform any further revisions that are needed.

### Study context
Data collection took place between December 2019 and December 2020. From March 2020, the UK enacted a number of restrictions in an effort to slow the spread of the COVID-19 (CV-19) virus. These ongoing measures included school closures, remote working where possible and social distancing restrictions and had a number of implications for our study. With much of the country moving towards remote learning and working during this time, many people became more familiar with using online technology[27 28] which likely facilitated engagement with our online screening questionnaires and intervention. Nonetheless, families and school staff had increasing and frequently changing demands on their time during this period, with parents being required to support their child's learning from home, often alongside working from home or managing other disruptions to their lives and teachers having to adapt and deliver lessons and support online as well as offering in-school teaching for some children. Schools had to respond to fluctuating school CV-19 regulations, while many staff were juggling their own caregiving responsibilities.

### Procedure modifications
The timing of the study, coinciding with UK CV-19 restrictions (March 2020 to December 2020), meant that some of our planned recruitment approaches and data collection strategies were altered, for example, face-to-face interviews had to be conducted via telephone/video call from March 2020. We had originally aimed to include interviews with parents who chose not to participate or dropped out of the intervention, as well as cued-recall interviews with 12 parents and 4 teachers about the experience of delivering or receiving feedback on screening questionnaire outcomes.[23] Because of the move to remote contact and because of the demands on teachers' time, we changed the procedure so that the study CWP provided feedback on screening outcomes to parents, rather than teachers. As such we did not interview teachers about their experience of delivering this feedback. Furthermore, we were unsuccessful in recruiting any non-participating parents and were only able to recruit a small number of parents who dropped out (n=2) and parents to cued-recall interviews (n=2). It is likely that CV-19-related demands on parent/school staff time and societal disruptions were contributing factors.

### Data analysis
During the codesign process, we made audio recordings of interviews and focus group discussions and photographed tabletop activities. Recordings were transcribed in full. Two approaches were taken for analysing the data: 'fast and direct' and 'slow and in-depth'. A description of the 'fast and direct' and 'slow and in-depth' analyses is provided below and in subsequent articles that drew on the data collected for transparency (Williamson *et al*, under review).

The 'fast and direct' approach involved the researchers making notes of the key findings during interviews, focus groups and from participants' comments on the generic pathway visual images in Stage 1. The key findings were collated and shared with the research team and dedicated stakeholder group and, where necessary, used to rapidly alter the research study procedures. For the 'slow and in-depth' approach, NVivo V.12 software was used to facilitate data analysis of interviews and focus groups. A template analysis approach was used.[29] This first required researchers to become familiar with the data by re-reading transcripts several times. The primary author (VW) then created a template of initial codes guided by the open-ended interview schedule questions, the empirical literature of child mental health and school-based interventions as well as the study's research questions.

In template analysis, the templates are study specific and the first iteration of any template in a given study provides the basis for further iterative developments. Once the template was developed, transcripts were analysed in a 'top down' manner following the provisional structure of the template. Data collection and analysis took place simultaneously to allow emerging topics of interest to be investigated further in subsequent interviews. Peer debriefing was carried out midway through data analysis and the template was modified to include additional codes based on discoveries in the dataset that

had not yet been captured by the initial coding template. Once all the data had been initially analysed, the populated templates were then shared, discussed and refined within the authorship team (CC, ML, TF, IM, VW, SHS, FM). Themes relating to the research question were identified in the coded dataset through analysis of patterns found between codes and among coded segments as well as through code use frequencies. Each theme was identified and verified through team consensus. Given that in this article we aim to provide a reflective and pragmatic account of the data, rather than providing an account organised by themes, we will focus on describing the challenges we faced throughout the codesign process at distinct research phases, the strategies we used to overcome these issues and reflections on the lessons we learnt, drawing on examples of previous codesign studies (see table 3).[18 30 31]

### Reporting and reflecting on experiences of codesign process findings

Based on the insights and outcomes from the codesign process, we present a snapshot of our findings related to the codesign and delivery of our school screening and intervention pathway for child anxiety problems (see table 3). We highlight the challenges faced by participants both prior to and during data collection structured by the patterns of participants' shared concerns in each research phase and steps taken to mitigate these difficulties. Our findings are organised by insights from the codesign process, are reported by distinct research phases and include data about how the pathway was experienced and perceived by users and influenced and adapted following their feedback. We present a simplified representation of the challenges, mitigations and lessons learnt in each research phase in table 3. Anonymised excerpts are provided to illustrate key points. The findings from the qualitative interviews and in-depth data analysis with practitioners and parents are reported in detail elsewhere (Williamson *et al*, under review).

### RESULTS
### Research phase: appraising the existing need for support and context

To successfully identify children with anxiety problems and facilitate access to early intervention, the pathway would need to overcome uncertainty about whether particular children are likely to benefit from intervention and create a clear route to access it. Previous studies[5 9 32] have shown that parents and teachers often struggle to identify whether the difficulties a child is exhibiting reflect a clinically significant problem. This was supported by data from our participating mental health practitioners who described that many families as well as school staff may not consider a child's emotional or behavioural difficulties as indicative of a likely problem, rather it may be seen as a 'phase' or attention seeking. As one practitioner describes:

Practitioner: You are aiming to reach out to parents that have never given a thought maybe that there [are] maybe anxiety issues in [their] children… I think some parents aren't aware at all and maybe quite oblivious to little tell-tale signs that might be going on and just to recognise it.

If this obstacle of identification was overcome and a child was recognised as having a likely anxiety problem, previous studies have found families may nonetheless be hesitant to engage in school screening due to concerns about the accessibility of formal support.[9] Participating practitioners and parents in the present study described the often extensive waiting lists for child and adolescent mental health services. Practitioners reported being overwhelmed by the demand for their psychological services and many families equally described being unable to promptly access appropriate formal support for their child. Readily accessible support was thus a key requirement of any developed screening/intervention pathway for participating parents, practitioners and school staff. This practitioner describes that a pathway would be well received given the significant challenges parents can face accessing care:

Practitioner: First and foremost I'd say that parents will be crying out for help. The children that I've worked with and our team…are crying out for help. It's one of the hardest things I've seen is when a parent wants their child to thrive, and they can't [get them help] …I'd say parents will bite your hands off.

### Research phase: engaging schools

Participating teachers and school staff in Stage 1 described that schools are often bombarded with offers for their school to receive mental health programmes. Such programmes were often described as costly with unclear efficacy. Moreover, particularly in light of the CV-19 pandemic, schools were described as being under increasing pressure to provide psychological support to children. To build school trust and confidence in a screening/intervention pathway, teaching staff described the need for a pathway to be seen as credible and evidence based, with recognisable logos on materials, clear information provided to staff about pathway procedures, with further information readily available on request. One teacher describes the challenges faced by schools and the importance of demonstrating credibility below:

Teacher: I literally get ten emails a day offering us some sort of mental health intervention… saying 'sign up for our pack, it's only £X thousand.'… That's the question isn't it, it's like how are you going to prove to schools…that actually this [pathway] is better than X, Y or Z?… I think credibility is really key with this…. Just because there's so much out there now. It's really hard as a teacher I think to make a value judgement.

**Table 3** Challenges, mitigations and lessons learnt from qualitative data collections

| Challenges encountered | How we mitigated these | Lessons learnt |
|---|---|---|
| **Research phase: appraising the existing need for support and context** | | |
| Parents/teachers may not recognise anxiety as a problem | Offering universal screening for the Y4 class. | Universal screening offered a way to identify children who may be struggling with anxiety, but difficulties were not previously recognised as such. |
| Parents may not know how to access help for their child | Integrated pathway for screening and intervention so families are offered help if potential difficulties were identified. | Schools and families were receptive to a screening programme if an intervention to problems found was also being offered. |
| Formal support may not be easily accessible | Integrated pathway included screening and intervention so families would not need to be referred elsewhere to access support for anxiety problems. Rapid contact with a mental health professional was available to support further signposting to resources and further services if required. Intervention was made available to all families interested in taking it up, not solely those who screened positive for a likely anxiety problem. | An inclusive offer for access to a low level intervention was of interest to families, even those who did not have a child who screened positive for a likely problem. Low level or early mental health interventions may not be sufficient for complex needs cases and team must be prepared to provide resources and make referrals as part of the intervention. |
| **Research phase: engaging schools** | | |
| School staff are bombarded with offers for mental health interventions | Used university logos on materials, refer to previous evidence, and offer face-to-face meetings with staff to answer questions. | Future studies should take steps to ensure school-based screening/intervention studies are seen as credible and trustworthy to schools. |
| Schools are under considerable and changing CV-19 pressures to provide children with mental healthcare | Pathway incorporates an efficient intervention to be offered to families in cases where children met criteria for likely anxiety problems and which can be delivered remotely. | There is an increasing demand for schools to offer accessible mental health support to children and young people due to CV-19 and a screening/intervention pathway may be especially welcome as a consequence. |
| **Research phase: participant recruitment** | | |
| There may be stigma around mental health problems and help seeking. There may be a lack of trust in formal services and interventions where families have had negative previous experiences | Universal screening was offered to Y4 within a supported information session at school. Information was shared with parents and school staff explaining all procedures, including guidance to address data sharing concerns. | 'Opt out' (rather than 'opt in') was considered to be a more inclusive approach for engaging families, that is, all children are included unless parents/carers request for them not to be. Parents/carers are given clear information and opportunities to 'opt out'. |
| Schools and families may not have a good understanding of mental health | Training materials were provided to staff about the project which included psychoeducation. Staff training briefing, including in-person meetings, telephone calls and a short information video was offered. Assembly, an in-class lesson and parent evenings were offered to provide psychoeducation to children and parents/carers. | Brief video about the pathway and the steps involved was considered more accessible and engaging than an information sheet. School staff reported not being approached by families to ask questions about the pathway but nonetheless staff appreciated being informed about how the pathway operated. Being able to contact the research team and receive personalised feedback was valued and allayed parents' concerns. |
| Parents did not attend information sessions or reported not hearing about the project | Brief information video about the project made and posted online and circulated via school mailing lists. | Delivery of information in a varied and accessible format (eg, information video) is preferred by parents who often have many competing demands on their time. |
| **Research phase: screening** | | |
| Concerns about the accuracy and content of screening questionnaires | Underpinning work to improve accuracy and content of screening measures (with stakeholder involvement). Clear information was provided to parents and teachers about the content and purpose of the questionnaires in advance. Parents had the option for their child to complete the questionnaire at home with them instead of in class. Screening for likely case criteria was done by encouraging parent, child and teacher completion of the screening questionnaires to provide a more complete picture of the child's difficulties. Language for communicating about screening developed with stakeholders to ensure sensitivity. | Researchers must be transparent and clear when giving information to families and school to ensure school-based screening/intervention studies are understood and are credible and trustworthy. It is important to stakeholders that multiple views about a child's anxiety are heard to reflect the different experiences in different contexts. |
| Schools feel unable to offer a screening session in classrooms | Dedicated team facilitate administration of screening questionnaire session in small groups outside the classroom. Information assembly and in-class lesson provided by research team to explain what the questionnaires were for in context of wider psychoeducation. | Having a dedicated team presence can feel reassuring to teachers who may lack confidence in having mental health-related discussions. This approach may also reduce burden for staff. |
| Concerns about adequate privacy during screening questionnaire completion | The option of completion of screening questionnaire at home via online/paper was also offered to children. Option to complete in classroom on a tablet was offered. | Participating children ultimately did not report privacy concerns if they completed the questionnaires at school (pre-CV-19). Children enjoyed taking part in the study and feeling 'part of' the pathway. Having the option for their child to complete at home was felt to be reassuring for parents. Tablet option was considered more engaging as well as ensuring privacy. |

Continued

**Table 3** Continued

| Challenges encountered | How we mitigated these | Lessons learnt |
|---|---|---|
| Concerns about the ability of families to take part when schools moved to remote learning due to CV-19 | Schools were provided with information sheets and envelopes to mail home to families as schools reported that families were inundated with emails and postal communication was preferred (although this was not taken up by families). Online questionnaires were delivered via a user-friendly and secure platform. The dedicated teams were available and responded quickly to teacher/parent questions about the study and accessing the questionnaires. | Responding to parental needs, such as being overwhelmed by emails and delivering information via other channels, helped to disseminate accessible information about the study. Responding quickly to concerns helped to continue families' and staff interest and trust in the project. Families found that due to increased remote working and school work, completion of online questionnaires for the study was not challenging and they did not have concerns about data being stored online. Postal response rate was low (during CV-19 restrictions). |
| **Research phase: feedback of screening outcomes** | | |
| School staff have considerable pre-existing demands on their time | Dedicated team delivers feedback to families about screening questionnaires directly. | Families found feedback from the CWP directly to be acceptable as the practitioner was seen as a neutral party, independent of the school, and could answer their queries. |
| Parents may find the feedback surprising or may be distressed to hear that their child has possible anxiety problems | Stakeholders gave input into the content of the feedback letter to families. This letter was followed up by a phone call to discuss any concerns and answer questions. | Feedback of screening questionnaire scores may be a shocking (or validating) moment for families and research teams should be prepared to approach the subject sensitively. |
| Parents of children who screen positive for likely anxiety problems may choose not to take up the intervention | Future help seeking is encouraged by making it clear that treatment is potentially accessible. Resources are provided which could be useful in future. A psychoeducation lesson is provided to all children including simple guidance on managing anxiety. | Future studies should consider what appropriate steps can be taken to support child anxiety problems where parents are not able to participate in the intervention for any reason. |
| Parents of children who screen positive for a likely problem may feel they are being forced to take up the intervention | Important to highlight that the intervention is optional and that the school/other services will not be informed whether or not they choose to be involved in the intervention. | It is essential that clear information is given about confidentiality (and its limits) and data sharing to reassure families. Researchers should be conscious and sensitive that not all families may have positive supportive relationships with their child's school/services. |
| School staff feel they should be informed about the children meeting criteria for potential anxiety problems to fulfil their duty of care | School staff are copied in to feedback letters that are sent to families where parents consented. | School will have procedures in place to fulfil their duty of care to children that must be considered when identifying potential child anxiety problems. |
| **Research phase: delivery of online intervention** | | |
| Parents feel they would benefit from peer support | This potential add on was explored with parents and what format this would be preferred given CV-19 social distancing restrictions (eg, WhatsApp, Facebook group). | Future studies should bear in mind the context in which parents engage with mental health interventions and that they may find informal peer support valuable for themselves as well. |
| Lack of school attendance due to CV-19 removed many sources of children's anxieties | Information highlighted that skills learnt in the parent intervention will be applicable for the future. Responses to routine parent questionnaires needed to be interpreted in the context of CV-19 circumstances (eg, children not attending school). | It is essential to be prepared to adapt or respond when measures are not applicable to the context. |
| Parents may not feel an online intervention is acceptable as opposed to more traditional face-to-face support | Families were informed that the intervention that was being delivered online was based on a widely used treatment. | Parents found the online intervention to be acceptable and it often fitted better around their schedules than face-to-face support. Weekly phone calls from the CWP were felt to be essential to personalise the experience and maintain momentum. |
| Parents are concerned about next steps to support their child once the intervention modules are completed | CWP highlighted that referrals would be made for further support if needed after the intervention. A phone call from well-being practitioner was delivered at 4-week follow-up to embed learning and offer guidance. | It will be important to be prepared to support making referrals on to other local services if the intervention offered does not entirely resolve child's difficulties. |
| **Research phase: assessing secondary impacts of pathway** | | |
| Concern that involvement in the study may lead to children being labelled or bullied | Clear information provided to teachers, children, and families via school assembly, in-class lesson and information sheets which includes psychoeducation about mental health. Confidentiality is explained to families, including what data will and will not be shared with the school. | It is important to be mindful that mental health stigma is an endemic issue but providing psychoeducation as part of the school-based screening/intervention represents an opportunity to improve language around and understanding of mental health. |
| Ensuring that the pathway maximises potential for wide and long-term benefits, for example, through increased mental health literacy in school context | Psychoeducation provided about mental health in several stages, including during teacher training about the project, parent information sheets and feedback, as well as during the assembly and class lesson for children. | There is the potential for school communities to have improved emotional and mental health literacy via the dissemination of linked psychoeducation. Future evaluations should aim to track changes over time in mental health stigma in schools—such as before and after study implementation—and tailor their psychoeducation and information sheets accordingly. |

CV-19, COVID-19; CWP, children's well-being practitioner; Y4, year 4.

## Research phase: engaging families

Once schools had agreed to be involved in the delivery of the pathway, Y4 children and their parents were invited to consent/assent to screening. Practitioners and teachers described that stigma-related concerns may prevent families from participating in this key step of the pathway,

preventing them from benefitting from early identification. This is consistent with the broader literature on barriers to help-seeking and illustrated by the following excerpts:

> Practitioner: Yes, it's convincing every parent that this [pathway] is good because some parents don't want a label or don't want to admit things. But the majority want to embrace it. Some parents will go 'no way!' and it could be that they are the ones that are flagged up.

> Teacher: Parents should be talking to us about if they're concerned. It shouldn't have to wait for this sort of intervention but often it does because families aren't always very good at that. Some families like to cover [up] these things and that's what you are aiming to unpick isn't it is where families like to downplay or deflect when there really are problems.

On the other hand, parents who had faced challenges previously in accessing formal help for their child reported that, as a result, their relationship with their child's school had sometimes become strained or they lacked confidence in formal psychological services/interventions. Concerns about the steps of the pathway, such as what data would be collected, from whom and whether they would be shared outside the research team, were frequently described by parents. The excerpt below illustrates the potential stigma-related concerns parents may have and how this could be mitigated by clear guidance:

> Parent: I guess the issue that some parents might have is where that information is going to be shared, there might be parents thinking 'oh I don't want a secondary school to know about, I don't want this to go on their records. I don't want them to be labelled in some way through this'…. I guess just [being] really explicit in the communication [to families] that this is just for your benefit, your child's benefit. It's not something that will label you or be recorded by school.

To overcome these participation concerns, several information sessions (eg, Y4 assembly, parents evenings, teacher briefings) were delivered to provide clear guidance about the pathway (including data sharing procedures), answer questions and allay concerns. As parents and staff had many demands on their time and some sessions were poorly attended, we made brief information videos and these were circulated among school staff and Y4 parents. Researchers also provided their contact details and encouraged staff/parents to get in touch with any further questions or concerns. Going forwards, it was also felt by practitioners, teaching staff and parents that an opt-out approach to screening (where all Y4 children are included unless parents request for them not to be), rather than the opt-in approach used, would make the pathway feel more inclusive and help overcome stigma-related barriers to participation. One parent described

how opt-out would still allow parents who were concerned to withdraw their children while providing most children the chance to participate:

> Parent: I think our daughter would have liked the opportunity to do [the questionnaire] and for someone to say 'that's OK, there isn't a right or wrong it's just about how you feel'… I think it should be part of the curriculum long term but… opt-out is the better option of what you have at the moment….Because if you feel really strongly, you still have that opportunity to pull your child out of it, but why you'd want to I just don't know.

### Research phase: screening

Once schools and parents had agreed to the delivery of the pathway, concerns were then encountered regarding the feasibility of delivering screening questionnaires for child anxiety problems in classroom settings. Parents in Stage 1 were concerned about the validity and content of the child screening questionnaires and whether child report was reliable. Whereas children participating in Stage 1 focus groups expressed concerns about whether there would be adequate privacy to fill in paper questionnaires in the classroom. Children were also concerned that sharing one's fears and worries may lead to negative outcomes, as one child describes:

> Child: Sometimes your worries can either be small worries which sometimes you can tell them but sometimes if they're big worries, like I've had some big worries before, I think you should probably just keep it to yourself….I would normally keep all my worries to myself because… if you keep it private then no one else is going to fiddle around with it and make it even worse.

In response to privacy concerns, the research team made it possible for the Stage 2 parent/child/teacher report screening questionnaires to be completed online using a secure platform (Qualtrics). Participating children and teachers in Stage 4 interviews ultimately did not describe experiencing concerns about classroom privacy. This early amendment was also especially opportune as it allowed families/staff to continue to participate from their homes when CV-19 restrictions and school closures later came into effect. Nonetheless, practitioners highlighted that some families may lack access to or confidence using online technology, and this may exclude some from participating.

To address parental concerns about the screening questionnaire content, we provided clear information about the content and purpose of the self-report questionnaires prior to consent. Parents were not routinely provided with a copy of their child's questionnaire responses, but researchers made a blank copy of the child-report questionnaire available on the study website so that there was transparency about questionnaire content. The triangulation of teacher, child and parent report was considered

by many participants to be a strength of the pathway as this thorough approach was seen as more reliable and comprehensive than a single point of view. One parent described the benefits of multiple reports below:

> Parent: As a teacher [myself], I used to feel very much that I knew things about my students that their parents didn't know because…I spent more waking hours with them than their parents did. And so I know your child, I can give you my observations confidently….I suppose an accurate picture of a child's disposition can't come from just one person because of the differences between being at home and school. So… I suppose I think that it's right that [the teacher] did [the teacher-report questionnaire] because anyone trying to help my daughter, if she needs help, needs to have as holistic a picture as possible.

Nonetheless, teachers stressed the many demands on their time and were concerned that they would not have capacity to deliver information about the screening and pathway to the class, support children in filling in their screening questionnaires as well as complete screening questionnaires on behalf of each participating child. In response to these concerns, the research team attended the school to deliver the information session, screening questionnaire administration and answer any questions. However, due to CV-19 restrictions, it was not possible for the research team to visit the third school in person. Where families completed the questionnaires remotely and had queries, teaching staff were encouraged to contact the research team who helped staff to draft replies. In Stage 4, teachers reported feeling that the questionnaires were easy to access, were not time consuming and research team presence for questionnaire administration was efficient and reassuring. As one teacher notes:

> Teacher: Yes, I think [taking part] didn't feel onerous in any way. I think is the upshot because so often again when you get embroiled in these things you realise that the paper filling and the time it takes is the thing that you hadn't anticipated. But [the pathway] didn't seem to take up any time at all in that sense…I didn't notice any issues with feedback, with admin or anything at all. So very positive from our perspective in that sense.

### Research phase: feeding back screening outcomes

Receiving feedback about the likelihood of a child meeting criteria for anxiety problems based on the screening scores was a key issue for many participants. Participating parents described that for some the news that their child had a likely anxiety problem was expected and feedback confirming this was reassuring. Other parents felt this feedback may be unexpected and distressing and may lead to feelings of self-blame or guilt. Practitioners highlighted the need for this feedback to be delivered sensitively and reassuringly to parents, with an emphasis on the availability of an evidence-based intervention. In response

to these concerns, the research team sought input from the dedicated stakeholder group into the contents of the feedback letter and a follow-up phone call with parents was also carried out to discuss any additional concerns or questions parents may have. This parent describes how they found receiving feedback to be a helpful, validating experience:

> Parent: I think we found the feedback really helpful. It was particularly helpful just because it felt like it validated some of the concerns that we have had… I think we just thought well…like no one is asking us how bad this is and so it must just not be that bad. So, to get the numbers back and to see oh our concerns are right, there are some numbers here that are quite alarming. I think we found that quite helpful.

Due to CV-19 school closures, the research team provided feedback to families directly via letter followed up with a telephone call. Stages 3 and 4 interviews with parents described feedback from the research team to be acceptable as researchers were seen as knowledgeable about child mental health and were also a neutral party, independent from the school—a feature that was particularly important if the family had had difficulties accessing support from the school in the past. This feeling is illustrated in the following excerpt:

> Parent: I think [the feedback is] better coming from you than from the school because you are not involved. I mean, I know you are involved, but you are not the teacher, you are not the headteacher, you are not the school cook, you are not to do with school…. Not one of the pupil's neighbours parents or something so you are neutral. I think it's better coming from you.

Research team feedback to parents directly, rather than school staff delivery, was also felt to protect families' privacy. On the other hand, school staff reported concerns that they had a duty of care to fulfil and should be informed which children met criteria for likely anxiety problems. To address both parties' concerns, where the parent consented, the research team provided schools with a copy of the feedback letter sent to each family, but staff were otherwise not informed whether a family chose to take up the intervention. Parents were also fully informed prior to participation about confidentiality and its limits, including what information would and would not be shared with the school by the research team.

In a similar vein, practitioners and school staff expressed concerns that some parents of children with likely anxiety problems may refuse the intervention or drop-out. These children were considered to be most vulnerable as well as most likely to benefit from the intervention. These parents were seen by some school staff and practitioners as uncaring or 'bad' parents, rather than as parents who were simply too overwhelmed to engage with the intervention or had had poor experiences of engaging with services in the past. This pattern of concern highlights

the balance that must be struck in a codesigned pathway between recognising and responding to varying stakeholder concerns while accepting that all participants have a right to refuse an intervention. Nonetheless, practitioners highlighted that sensitive delivery of screening feedback and a positively framed offer of optional formal support may increase future help-seeking even among parents who refuse the pathway intervention.

> Practitioner: I think there's something about the message of help isn't it and being able to provide a nice experience of accepting help or not accepting help so that when the family is ready or maybe when the child is old enough to opt-in on their own that they'll still have that positive memory.

### Research phase: delivery of online intervention

Participating parents highlighted that the CV-19 context influenced their experience of the online intervention that was offered as part of the pathway. For example, many parents reported being more comfortable working remotely and the online intervention was, therefore, seen as more acceptable and accessible. The weekly phone calls from the CWP were also felt by parents to be an essential part of the intervention process, personalising their experience of the online modules and maintaining their family's engagement with the modules. Nonetheless, the CV-19 social distancing restrictions meant many parents reported not having the opportunity to speak with friends or school staff informally about their experience of the pathway. The adjunct of social support, such as via a closed peer support group for parents, was considered to be a valuable component to consider in future studies, as this parent describes:

> Parent: I think the creation of a group would definitely help some people…I think there are people that would like to have those conversations within a safe space… and you know that other families are having…experiences that aren't too dissimilar to you and having that just makes it a bit more relaxing and it gives you the opportunity to open up about certain things. I think it's helpful to relax the worries that perhaps parents can have and it's not always your fault and it's not always what you are doing it's sometimes just having that openness just makes it easier.

Consistent with previous studies,[9] parents described concerns about the availability of follow-on support and how they would manage any residual anxiety problems their child may have once they completed the intervention. Similarly, several professionals expressed concerns about how families who were still struggling despite completing the pathway would be adequately supported. Nonetheless, this finding underscores the importance of having steps in place to support families beyond the intervention stage for the screening/intervention pathway to be considered acceptable. A core component of the present intervention pathway was to teach parents skills and strategies to support their child beyond the intervention. Moreover, a preplanned component of the intervention was for the study CWP to contact families 1 month post intervention to check in, and the content of the check in call was amended to ensure troubleshooting could be carried out as well as making referrals to further formal support where necessary.

> Parent: I think the fact is that even though you are discharging [families] if you identify that they need more help then you are going to point them in the right direction, aren't you? So, they aren't just being left in limbo which is important.

### Research phase: identifying and addressing potential secondary impacts of pathway

The pathway was generally perceived and experienced as a positive and helpful opportunity for families to support their child with anxiety problems. However, concerns were expressed that the delivery of a screening/intervention pathway in schools could cause some children to be labelled or bullied. Some practitioners felt this could be due to poor mental health literacy within schools, while parents described that bullying or labelling could arise if their data, such as whether their family were involved in the intervention, were shared across school staff. Nonetheless, the introduction of the pathway to a school was considered by parents, teachers and practitioners to be an opportunity to improve a school community's understanding of mental health. The research team acted on these insights by providing clear information about confidentiality as well as psychoeducation at several stages throughout the pathway, including during parent and teacher briefings and within the information sheets. The research team also delivered an in-class lesson focusing on psychoeducation about anxiety problems and problem solving to Y4 children following the screening session. A reduction in mental health-related stigma in schools is a frequently cited benefit of school screening/interventions.[33] Whether stigma is reduced in primary school settings following the implementation of such a pathway has yet to be evaluated but is an important direction for future research.

> Practitioner: I would hope that it would reduce the stigma around it and I would hope that it would be something that other parents would be interested in finding out more about and that as those children progress through school they can take what they've learnt with their parents and use it so that when they get to secondary school… to prevent it from being such an issue then.

In table 3, we present each research phase and detail the challenges, mitigations and lessons learnt in each phase informed by the codesign process.

### DISCUSSION

Using codesign and data collection from multiple sources, we identified several key barriers and facilitators

to participation for both schools and families, including difficulties accessing (or delivering) reliable mental health support for children; concerns about mental health-related stigma; concerns about the trustworthiness and effectiveness of the pathway; and the adverse impact of CV-19 restrictions on participation. Our iterative codesign approach allowed for the research team to actively respond to users' concerns which may have ultimately improved how the pathway procedures were experienced. As described in table 3, the developed pathway ultimately consisted of: (1) the circulation of credible and transparent study information and psychoeducation in a variety of formats to school staff, children and parents; (2) screening for anxiety problems using child, teacher and parent report online and paper questionnaires; (3) the sensitive delivery of written and verbal feedback to parents directly regarding screening questionnaire outcomes; and (4) the offer and delivery of a brief parent-led intervention.

### Recommendations for future school screening/intervention studies

Our findings offer key lessons for future studies aiming to deliver engaging and sustainable school-based screening and intervention procedures. For example, our study demonstrated that despite recent studies which have found that parents are the most effective reporters to identify anxiety diagnoses among preadolescent children,[20 34] our participants considered that the inclusion and triangulation of parents/child/teacher report on screening questionnaires was valuable. This highlights that future studies may need to strike a balance between what is psychometrically reliable and what procedures feel valid and meaningful to participants themselves to bolster engagement. Moreover, we found that parents were especially concerned about data privacy and sharing—particularly if they had previously had negative experiences with their child's school or formal services. The need to share participant data with school staff in order to meet their duty of care had to be carefully weighed against parents' concerns about them or their child being labelled or judged and a desire for privacy. In response, the research team opted for transparency, providing parents with clear information about what data would (and would not) be shared with whom, with consent obtained for this at the outset. CV-19 restrictions meant that the research team provided screening feedback to parents directly about the screening outcomes and this improvised solution was found to be preferable to families to feedback being given by school staff. Future screening/intervention efforts may benefit from using an independent source (eg, not connected to the school) who is knowledgeable about child mental health to deliver feedback to parents. Furthermore, receiving feedback on screening outcomes was found to be a crucial part of the pathway which if done well, could facilitate engagement with the intervention and/or encourage future help-seeking. Knowledge of which research phase(s) and

elements of the screening/intervention pathway may be especially critical—and produce potential long term positive outcomes—for participants may help to guide future studies. Taken together, these points underscore the need for evaluations to include consideration of the implications of procedures, involving stakeholders and users in actively considering what broader (and perhaps unexpected) outcome the steps taken may have.

### Merits and challenges of using a codesign methodology

In the present study, our use of codesign presented a number of benefits and challenges, as well as transferrable learning points that may be applicable to other studies. A core strength of using a codesign approach is that it allows for the recognition that users may have a variety of pre-existing and conflicting beliefs and concerns about mental health and help-seeking and ensures that these concerns are heard and can be effectively responded to.[12] In the present study, we were able to gain an in-depth understanding about what barriers and facilitators for pathway engagement exist and to cocreate solutions with our participants. For example, stigma-related concerns were expressed regarding the screening process which led to the recommendation that an 'opt-out' approach may be more inclusive. Our 'fast and direct' analytic approach meant the pathway procedures could be quickly and meaningfully adapted in response to feedback to help ensure optimal user engagement. The codesign methodology used also allowed for the collection of data from a broad range of users (parents, children, teachers, practitioners) at various stages of the pathway, providing in-depth insight into their experiences and concerns at each research stage. The inclusion of a range of perspectives highlighted that some school staff and practitioners may have very different views from families about the potential risks and benefits to a screening/intervention pathway. For example, a number of school staff and practitioners expressed beliefs that a screening process was beneficial as some parents may downplay or deflect child anxiety difficulties, while parents described school staff dismissing their concerns. Incorporating multiple views via codesign paints a fuller picture of the context in which a screening/intervention pathway is being introduced and can allow for key contextual factors to be recognised and considered. The inclusion of stakeholders as members of the research team also provided valuable guidance in shaping the initial 'blueprint' of the screening/intervention procedures which were further refined in subsequent focus groups and interviews. However, this inclusive approach to data collection did yield a considerable amount of data which could be challenging to manage and meaningfully report. Given the amount of research data that goes unpublished (or 'research waste'[35]), this is a consideration for future studies.

### Strengths and limitations

This study has several strengths. Among the strengths is the inclusion of key stakeholders in the research team

who not only provided guidance on procedures but also supported the development of sensitive participant-facing documents and interpretation of data. A second strength is the range of participant views included using multiple eliciting techniques and different time points, allowing concerns to be well captured and responded to. Third, the research teams were able to adapt to the unforeseen social distancing restrictions imposed following CV-19—for example, by carrying out data collection remotely—and our findings and adaptations may be useful to future studies that are likely to face similar difficulties for the foreseeable future. However, given the changes that were made, it is unclear how our adapted pathway procedures would be received by schools and families in 'normal' circumstances. That said, CV-19 has led to an increased demand for child mental health services[36] and the screening/intervention pathway procedures that have been developed here may ultimately have a beneficial impact in improving child mental health and delivering support to families through schools. Another strength of this study was the inclusion of schools in Stages 2–4 that had varying numbers of children with special educational needs and relatively high numbers of children with English as an additional language (who may generally be under-represented in research).

A number of weaknesses should also be highlighted. Schools with high numbers of pupils eligible for free school meals due to low family incomes were under-represented.[26] Despite the targeted recruitment of parents in challenging circumstances (eg, foster parents, military connected parents), another weakness is that our sample may not capture the diverse views of families with different backgrounds and who are living in different circumstances. The majority of participating adults (ie, parents, practitioners, school staff) in this study were also female which may limit the generalisability of the findings to fathers and male staff/practitioners. Future studies should endeavour to capture their views which are often overlooked in investigations of the development and treatment of anxiety disorders in children.[37] Moreover, possibly due to families being overwhelmed or difficult to contact due to CV-19 restrictions, we were unable to meet some of our recruitment targets (eg, for parents who chose not to participate in the pathway). Thus, a final weakness of this study is that comparatively little is known about why some families may chose not to take participate in the pathway and, as many of these families are likely to be those who could benefit the most, it is important that researchers establish how best to capture their perspectives in future research.

## CONCLUSIONS

Despite these limitations, this study adds to the literature in several ways. First, it illustrates that a screening/intervention pathway for child mental health problems in schools can be inclusively codesigned in partnership with parents, children, school staff and mental health practitioners. Given the sensitive and often stigmatised nature of mental health screening and treatment, this study highlights that a methodological approach such as codesign can lead to an in-depth understanding of users concerns and the cocreation of solutions, optimising study procedures and improving the chances of successful implementation. A well-designed screening/intervention pathway may bridge the gap between children and families' needs for and access to early mental health treatment which is pressingly required given the extensive waiting lists and high thresholds for accepting referrals for many specialist services.[38] Finally, the findings from this study underscore that there may be tangible potential secondary benefits to offering a well-designed school-based screening/intervention pathway. An effective and acceptable pathway could not only foster child well-being but also promote future help-seeking, highlighting that school-based screening/intervention efforts for child mental health are both promising and worthwhile. Future studies should systematically evaluate the codesigned pathway to examine whether reductions in child mental health problems are achieved and if wider benefits are found.

**Author affiliations**
[1]Institute of Psychiatry, Psychology and Neuroscience, King's College London, London, UK
[2]Department of Experimental Psychology, Anna Watts Building, University of Oxford, Oxford, UK
[3]School of Psychology & Clinical Language Sciences, University of Reading, Reading, UK
[4]Institute for Neurodevelopment and Health, Department of Psychology, Aston University, Birmingham, UK
[5]University of Bath, Claverton Down, Bath, UK
[6]Australian Institute of Suicide Research and Prevention and School of Applied Psychology, Griffith University, Mount Gravatt, Queensland, Australia
[7]Thames Valley Clinical Trials Unit, University of Reading, Reading, UK
[8]Charlie Waller Memorial Trust, Thatcham, UK
[9]NIHR ARC South West Peninsula, University of Exeter, Heavitree Rd, Exeter, UK
[10]Department of Psychiatry, University of Cambridge, Cambridge, UK
[11]Nuffield Department of Population Health, University of Oxford, Oxford, UK
[12]Population Health Science Institute, Faculty of Medical Sciences, Newcastle University, Newcastle upon Tyne, UK
[13]Stanley Primary School, Teddington, UK
[14]Bransgore C Of E Primary School, Bransgore, UK
[15]West Berkshire Council, Council Offices, Newbury, UK
[16]Square Peg (Team Square Peg CIC), London, UK

**Acknowledgements** The authors would like to thank those who participated in the patient and public involvement activities for their contribution to this research.

**Contributors** All authors contributed to the study design, contributed towards the write up of the manuscript and read and approved the manuscript prior to submission. VW, ML, TR, CC, RB, IG and SP contributed towards data collection and data analysis.

**Funding** This paper represents independent research funded by the National Institute for Health Research (NIHR) (CC) and hosted by Oxford Health NHS Foundation Trust. CC and MV acknowledge support from the Oxford and Thames Valley NIHR Applied Research Collaboration (ARC). MV was partly supported by the NIHR Oxford Biomedical Research Centre. OU was supported by the NIHR ARC for Leadership in Applied Health Research and Care for the South West Peninsula at the Royal Devon and Exeter NHS Foundation Trust. The views expressed are those of the authors and not necessarily those of the NHS, the NIHR or the Department of Health and Social Care.

**Competing interests** None declared.

**Patient and public involvement** Patients and/or the public were involved in the design, or conduct, or reporting, or dissemination plans of this research. Refer to the Method section for further details.

**Patient consent for publication** Consent obtained directly from patient(s).

**Ethics approval** This study involves human participants and was approved by Central University Research Ethics Committee at the University of Oxford (REF R64620/RE001). All adult participants gave written informed consent and children assented to participate in the project.

**Provenance and peer review** Not commissioned; externally peer reviewed.

**Data availability statement** No data are available.

**ORCID iDs**
Victoria Williamson http://orcid.org/0000-0002-3110-9856
Obioha Ukoumunne http://orcid.org/0000-0002-0551-9157
Alastair Gray http://orcid.org/0000-0003-0239-7278

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
