## [Reviewer comments · BMJ Open]

ARTICLE DETAILS

TITLE (PROVISIONAL)	School-based screening for childhood anxiety problems and intervention delivery: A co-design approach
AUTHORS	Williamson, Victoria; Larkin, Michael; Reardon, Tessa; Pearcey, Samantha; Button, Roberta; Green, Iheoma; Hill, Claire; Stallard, Paul; Spence, Susan; Breen, Maria; McDonald, Ian; Ukoumunne, Obioha; Ford, Tamsin; Violato, Mara; Sniehotta, Falko; Stainer, Jason; Gray, Alastair; Brown, Paul; Sancho, Michelle; Morgan, Fan; Jasper, Bec; Creswell, Cathy

VERSION 1 – REVIEW

REVIEWER	Rimvall, Martin Child and Adolescent Mental Health Services, Mental Health Services, Capital Region of Denmark, Research unit
REVIEW RETURNED	12-Nov-2021

GENERAL COMMENTS	Thank you for the opportunity to review this manuscript. In my opinion it generally reads well, and some aspects have actually inspired me personally for designing future psychotherapy interventions in the future. I only have relatively minor comments which I think should be pretty easy to look into. I hope they are helpful to the authors. Abstract: For international readers, it would be nice if the authors stated the age of "year 4" children already in the abstract. Introduction: The introduction is simply well-written in my opinion. It is rather lengthy, but really does provide a strong rationale behind the study, and properly explains the idea of co-design for non-expert readers. Methods: Stakeholders and PPI: It is confusing that this is described at the start of the methods before the recruitment description? Also, results from this part are described prematurely in the methods section in my opinion. Page 13: (CWP, NHS Band 5) – NHS Band 5 - this is not understandable for international readers. Also, what is the general role/education of a child well-being practitioner. Page 15: Two parents with "lived experience" were included in the PPI and stakeholder group. Just for clarification; lived experiences of anxiety themselves, or lived experience as a parent of a child with anxiety? Table 1: What is "SEN support"? This abbreviation should be written out/explained.
---

	Page 16: It is unclear to me exactly how the participants for this study were selected from the different schools. In Table 1 some basic school demographic characteristics are stated; but do these characteristics follow through to the actual participants (Table 2) – this is important with regard to representability. Page 17: Formatting mistake? After table 1 there is simply a reference out of the blue; "years (26)". Page 28-29: Would it perhaps be interesting to publish the initial and final templates in the supplement? This would be helpful with regard to replicability. Perhaps the same with the schedules for the open-ended interviews? Results Alluding to the above comment about findings from stakeholders/PPI, I think it would be appropriate to summarize the findings/lessons learned at the start of the results section. I think the results section is very well written and surprisingly concise considering the amount of data. Table 3 is particularly nice for the busy reader that does not have time to read the more elaborated citations from the participants etc. It provides many helpful tips for practitioners/researchers in the "lessons learned" section. Discussion I follow all the major points made by the authors. I think the discussion would be much easier to follow with some sub-headings in my opinion, if that is possible? It's easy to get a bit lost. Perhaps the summary of the findings can be shortened, when they are not directly related to a theme for discussion? E.g. lines 30-50 on page 50 removed? Limitations section: As it currently stands, this section jumps back and forth between limitations and strengths. It would be more transparent if they are listed separately. Limitations: Perhaps it should be noted that the sex distribution among participants is rather skewed. Any reflections on how that might affect the results/generalizability? I wish the authors good luck with their work!
--	--

REVIEWER	Lubans, David University of Newcastle, School of Education
REVIEW RETURNED	03-Feb-2022

GENERAL COMMENTS	General comments The aim of this study was to describe the process of a co-designed screening and intervention pathway for children with likely anxiety problems in primary school settings. The authors should be commended on their rigorous research design (i.e., multiple research stages, co-design approach) which allowed input from a range of stakeholders (i.e., students, teachers, parents, and practitioners). The study addresses an important issue, and the manuscript is well written. I have only minor comments for the authors to consider. Abstract I'm not sure if 'co-design' is an accurate description of the study design. Please elaborate.
---

	The results section does not describe any findings from the study (it somewhat reflects the study methods). I'm aware this is challenging considering the breadth of the results section, however, perhaps the authors could highlight some key findings or trends/themes for this section of the abstract. Methods Stage 4: It is not immediately clear (in text) how parent data collected during this stage were different from data collected in stage 3 (which is specific to experiences of receiving feedback on child's screening outcomes). Can the authors provide more concise detail regarding data that were collected from parents during stage 4? Procedure modifications: "We had originally aimed to include interviews with parents who chose not to participate or dropped out of the study". Should this refer to the 'intervention' as rather than the study? Table 2: Is there a specific reason that students' demographic information (age and sex) is not provided for participants in stage 2? Results Can the authors provide a summary of all participant characteristics (regardless of research stage) in text? This information would be useful to provide a general overview of study participants. The presentation of the results section is somewhat difficult to follow as it does not align clearly with the stages of research that are discussed in the methods section. Discussion The discussion could benefit from further interpretation of study results/findings. At present, the discussion focuses heavily on the study design/methods rather than study findings. It would be informative if the authors could highlight the main challenges that were experienced, and potential solutions or recommendations for the design of future screening/intervention pathways. Figure 1 I suggest including the number of participants that data were collected from at each timepoint/research phase. I'm aware this is already displayed in table 2, however organising this information by research stage (e.g., stage 1-4), as opposed to participant grouping (e.g., practitioner, parent, teacher, student) may be easier to interpret.
--	---

VERSION 1 – AUTHOR RESPONSE

Reviewer: 1

For international readers, it would be nice if the authors stated the age of "year 4" children already in the abstract.

We have added this to the abstract.

Stakeholders and PPI: It is confusing that this is described at the start of the methods before the recruitment description? Also, results from this part are described prematurely in the methods section in my opinion.

We deliberately introduced the Stakeholder and PPI early in the methods section as collaboration with these groups underpinned all the work that follows. We state on page 8 that a number of pre-determined study stages were informed by the stakeholder group to ensure that readers are clear how the research team came to decide on these stages. In line with standard guidance for PPI we don't consider this group to be 'research participants', and their input does not constitute 'research data'; rather this group worked alongside or as members of the research team. We hope that this makes clear why the contents of this section have been placed where they are.

Page 13: (CWP, NHS Band 5) – NHS Band 5 - this is not understandable for international readers. Also, what is the general role/education of a child well-being practitioner.

We have now edited to clarify that Children's WellBeing Practitioners (CWPs) are psychological therapists with a 1 year postgraduate training and have added a reference to NHS Band 5 for interested readers to improve clarity on page 8. This reference includes guidance about the role and training of a CWP.

Page 15: Two parents with "lived experience" were included in the PPI and stakeholder group. Just for clarification; lived experiences of anxiety themselves, or lived experience as a parent of a child with anxiety?

We have clarified that these are parents of a child with anxiety problems on page 9:

This group included two parents with relevant lived experience as a parent of a child with anxiety problems, two school leaders, and one school mental health lead for a national charity

Table 1: What is "SEN support"? This abbreviation should be written out/explained.

We have clarified SEN in Table 1.

Page 16: It is unclear to me exactly how the participants for this study were selected from the different schools.

We would be happy to provide further information. However, we are uncertain what aspect the reviewer is unclear about. On page 11-12 we state in detail how the different participants in different stages were selected:

Parents and children: To recruit participants with a broad range of perspectives to Stage 1, we circulated study invitations to families of all Y4 children in two primary schools in the local area, as well as circulating study adverts online on social media, and national mailing lists. In Stages 2-4, study information was circulated to all Y4 parents and children in three participating schools, including invitations to take part in the screening/intervention pathway and the opportunity to participate in study-related interviews. All Y4 parents and children in participating schools were invited to participate and were included in the study if they provided informed consent/assent.

Notably, in Stage 4, we also specifically recruited a number of parents facing challenging circumstances that could influence their views of the acceptability of and likely engagement with a school-based screening and intervention programme. These were parents who care for a foster child or a child with chronic physical health problems, where the parent has past/present mental health problem(s), or where the parent is a member of the UK Armed Forces community. This sub-group of parents (n=10, see Table 2) was recruited via circulation of study advertisements online and via mailing lists. Parents who expressed an interest in taking part were approached by the research team, screened against study inclusion/exclusion criteria, and invited to take part following informed consent. The inclusion of this sub-group of parents aimed to ensure that the co-designed school-based programme would be inclusive and appropriate to the needs of a greater number of families (see Williamson et al., under review).

School staff and other stakeholder participants. To recruit school staff and practitioners that provide mental health support in schools to Stage 1 and 4, we circulated invitations for study interviews/focus groups within local primary schools and shared study adverts online and via mailing lists. School staff and practitioners were encouraged to contact the research team if they were interested in taking part. School staff were included in study interviews if they were employed in a participating mainstream primary/junior school in England (e.g. class teacher, headteacher). The inclusion criteria for staff that provide mental health support in schools were that they must be a practitioner providing mental health support in primary schools in England, such as educational psychologists, Special Educational Needs Coordinator (SENCOs), and Emotional Literacy Support Assistants (ELSAs) (Stages 1 and 4, see Figure 1 and Figure 2). For clarity, they are referred to throughout this manuscript as 'practitioners'. Practitioners were sampled to ensure that a range of views were represented from a diverse group of professional backgrounds and qualifications.

In Table 1 some basic school demographic characteristics are stated; but do these characteristics follow through to the actual participants (Table 2) – this is important with regard to representability.

We have added further demographic information about the parents and children participating in stage 2 in Table 2. No children in the present study were reportedly eligible for free school meals. We did not collect information as to whether the child had SEN or had English as a second language.

Page 17: Formatting mistake? After table 1 there is simply a reference out of the blue; "years (26)".

This seems to have been a formatting problem on submission as 'years' refers to a footnote of the Table 1: National average = refers to official UK Gov statistics for the 2020/2021 school year [27].

Page 28-29: Would it perhaps be interesting to publish the initial and final templates in the supplement? This would be helpful with regard to replicability. Perhaps the same with the schedules for the open-ended interviews?

In terms of replicability, in template analysis, the templates are developed over several iterations and are study specific so while we'd be happy to provide them for transparency, they may not be useful to other researchers aiming to carry out similar studies. We have added a statement on page 19 detailing the development of the study specific templates below. Regarding the interview schedules, we are happy to add our interview schedules to supplementary material.

The primary author (VW) then created a template of initial codes guided by the open-ended interview schedule questions, the empirical literature of child mental health and school-based interventions as well as the study's research questions.

In Template Analysis, the templates are study specific and the first iteration of any template in a given study provides the basis for further iterative developments.

Alluding to the above comment about findings from stakeholders/PPI, I think it would be appropriate to summarize the findings/lessons learned at the start of the results section.

We have given this suggestion considerable thought as described above. As BMJ Open recommends a PPI section in the methods section, we feel that the lessons learned from the PPI group fits most clearly in the methods section to avoid confusion for readers. We are very happy to take further instruction from the Editor on this point.

I follow all the major points made by the authors. I think the discussion would be much easier to follow with some sub-headings in my opinion, if that is possible? It's easy to get a bit lost.

We have added sub-headings to the discussion.

Perhaps the summary of the findings can be shortened, when they are not directly related to a theme for discussion? E.g. lines 30-50 on page 50 removed?

We thank the reviewer for the suggestion however we are not sure which lines they are referring to as the paper is only 40 pages long. However we have taken care to ensure that the discussion is written concisely while covering the key considerations.

Limitations section: As it currently stands, this section jumps back and forth between limitations and strengths. It would be more transparent if they are listed separately.

On page 39 & 40 we have amended our strengths and limitations section to more clearly describe the strengths and weaknesses of this study:

A number of weaknesses should also be highlighted. Schools with high numbers of pupils eligible for free school meals due to low family incomes were underrepresented [26]. Despite the targeted recruitment of parents in challenging circumstances (e.g. foster parents, military connected parents), another weakness is that our sample may not capture the diverse views of families with different backgrounds and who are living in different circumstances. The majority of participating adults (ie parents, practitioners, school staff) in this study were also female which may limit the generalizability of the findings to fathers and male staff/practitioners. Future studies should endeavour to capture their views which are often overlooked in investigations of the development, and treatment of anxiety disorders in children [38]. Moreover, possibly due to families being overwhelmed or difficult to contact due to CV-19 restrictions, we were unable to meet some of our recruitment targets (e.g. for parents who chose not to participate in the pathway). Thus, a final weakness of this study is that comparatively little is known about why some families may chose not to take participate in the pathway and, as many of these families are likely to be those who could benefit the most, it is important that researchers establish how best to capture their perspectives in future research.

Limitations: Perhaps it should be noted that the sex distribution among participants is rather skewed. Any reflections on how that might affect the results/generalizability?

The reviewer is right to point out that the majority of participating adults in this study were female and we have reflected on the impact this may have on page 39.

The majority of participating adults (ie parents, practitioners, school staff) in this study were also female which may limit the generalizability of the findings to fathers and male staff/practitioners.

Future studies should endeavour to capture their views which are often overlooked in investigations of the development and treatment of anxiety disorders in children [38].

Reviewer: 2

Abstract: I'm not sure if 'co-design' is an accurate description of the study design. Please elaborate.

Given the word limits of the abstract, we have not added a statement to the abstract about what 'co-design' consists of however we provide a detailed description in the introduction on page 6 & 7.

For such a programme to be implemented, it must function efficiently, be safe and reliable, and have the experiences of service users and stakeholders at the heart of programme design and delivery [11]. This final criterion is best met by co-design - a method which aims to develop a thorough understanding of how stakeholders and service users perceive and experience the look, feel, and procedures of a service which is then used to inform the design and delivery of, and adaptations to, services [12]. This approach brings advantages over surveys or questionnaires of patient/stakeholder experiences of a service as it allows for an in-depth understanding of a service's potential shortcoming and/or the development of solutions. A co-design approach allows for both participant views as well as patient and public involvement (PPI) perspectives to be incorporated, ensuring services are designed for users with users [13]. Co-design has been widely used in health contexts to make services more acceptable and, thus, ultimately improve patient wellbeing (e.g. [14–16]). In relation to designing and delivering mental health services for children, previous qualitative co-design studies have yielded promising findings when the views of children, family members, clinicians, and other stakeholders were incorporated [17–19]. Designing and implementing a successful school-based screening and intervention programme for childhood anxiety disorders requires equally thorough triangulation.

The results section does not describe any findings from the study (it somewhat reflects the study methods). I'm aware this is challenging considering the breadth of the results section, however, perhaps the authors could highlight some key findings or trends/themes for this section of the abstract.

We state on page 20 that the aim of this paper is to provide an account of the co-design process, rather than themes.

Given that in this article we aim to provide a reflective and pragmatic account of the data, rather than providing an account organised by themes, we will focus on describing the challenges we faced throughout the co-design process at distinct research phases, the strategies we used to overcome these issues and reflections on the lessons we learnt, drawing on examples of previous co-design studies (e.g. [18,31,32]).

Methods - Stage 4: It is not immediately clear (in text) how parent data collected during this stage were different from data collected in stage 3 (which is specific to experiences of receiving feedback on child's screening outcomes). Can the authors provide more concise detail regarding data that were collected from parents during stage 4?

We have clarified on page 14 that parents interviewed in Stage 4 differed from the interviews conducted in Stage 3 as they were asked to reflect on their experiences of participating in the study as a whole (and were not invited to listen to and focus on the parent-CWP call):

Parent interviews differed from the cued-recall interviews (Stage 3) in that parents were asked about their overall experience of the screening/intervention.

Procedure modifications: “We had originally aimed to include interviews with parents who chose not to participate or dropped out of the study”. Should this refer to the ‘intervention’ as rather than the study?

Yes. We have amended this statement in line with the reviewer recommendation on page 15:

We had originally aimed to include interviews with parents who chose not to participate or dropped out of the intervention, as well as cued recall interviews with 12 parents and four teachers about the experience of delivering or receiving feedback on screening questionnaire outcomes

Table 2: Is there a specific reason that students’ demographic information (age and sex) is not provided for participants in stage 2?

We thank the reviewer for highlighting this point and we have added this demographic information to Table 2.

Results - Can the authors provide a summary of all participant characteristics (regardless of research stage) in text? This information would be useful to provide a general overview of study participants.

We have reflected carefully on this point. Given that participant demographic characteristics are thoroughly described in the method and Tables, we feel that adding another statement about the demographic characteristics would be repetitive and detract from the study findings. However, we have reflected on areas where certain groups are underrepresented in the discussion to highlight, for example, the high proportion of female parents and school staff (page 39).

The presentation of the results section is somewhat difficult to follow as it does not align clearly with the stages of research that are discussed in the methods section.

We have considered this recommendation carefully as Reviewer 1 found the results section to be concise, informative and clear and we are somewhat uncertain why Reviewer 2 experienced these difficulties. We state on page 20 how our findings are presented by distinct research phase and we summarise the findings by research phase in Table 3. We are, of course, happy to take instruction from the Editor on this point and make suggested alterations to improve clarity for readers.

The discussion could benefit from further interpretation of study results/findings. At present, the discussion focuses heavily on the study design/methods rather than study findings. It would be informative if the authors could highlight the main challenges that were experienced, and potential solutions or recommendations for the design of future screening/intervention pathways.

We thank the reviewer for this suggestion. In line with reviewer 1’s recommendations, we have added sub-headings to the discussion which we hope will better guide the reader about what recommendations can be made to future studies based on our findings as well as the merits and challenges and solutions of conducting co-design and wider study strengths and limitations.

Figure 1 - I suggest including the number of participants that data were collected from at each timepoint/research phase. I’m aware this is already displayed in table 2, however organising this information by research stage (e.g., stage 1-4), as opposed to participant grouping (e.g., practitioner, parent, teacher, student) may be easier to interpret.

We have added the sample sizes to the Figure as suggested and defer to the Editor to decide which figure version is preferred for publication. The revised figure is below:

VERSION 2 – REVIEW

REVIEWER	Rimvall, Martin Child and Adolescent Mental Health Services, Mental Health Services, Capital Region of Denmark, Research unit
REVIEW RETURNED	08-May-2022

GENERAL COMMENTS	I have no further comments As to the issue regarding the presentation of the findings from the PPI/stakeholder interviews, I fully accept that this is an editorial decision. Regarding the supplementary details provided by the authors about the selection of the participants were iformative, thank you. I apologize that my comment was not so clearly formulated. Again, I wish the authors good luck with their work.
---

REVIEWER	Lubans, David University of Newcastle, School of Education
REVIEW RETURNED	28-Apr-2022

GENERAL COMMENTS	The authors have been very responsive to the reviewers' comments and the revised manuscript is much improved. I believe that each comment was carefully addressed and the authors used appropriate judgment in deciding whether to alter the text or not.
--